# Treatment of Peri-Implant Mucositis with Repeated Application of Chlorhexidine Chips or Gel during Supportive Therapy—A Randomized Clinical Trial

**DOI:** 10.3390/dj7040115

**Published:** 2019-12-11

**Authors:** Philipp Sahrmann, Cyrill Bettschart, Daniel B. Wiedemeier, Ahmed Al-Majid, Thomas Attin, Patrick R. Schmidlin

**Affiliations:** 1Clinic for Preventive Dentistry, Periodontology and Cariology, Center of Dental Medicine, University of Zurich, CH 8032 Zurich, Switzerland; ahmedalmajid25@gmail.com (A.A.-M.); thomas.attin@zzm.uzh.ch (T.A.); patrick.schmidlin@zzm.uzh.ch (P.R.S.); 2Alte Gasse 24, 6438 Ibach SZ, Switzerland; cyrill17@hotmail.com; 3Center of Dental Medicine, University of Zurich, CH 8032 Zurich, Switzerland; daniel.wiedemeier@zzm.uzh.ch

**Keywords:** peri-implantitis, chlorhexidine, local anti-infective agents, maintenance

## Abstract

**Running head:**

Peri-implant maintenance with CHX

**Abstract:**

**Background:** To assess the effect of chlorhexidine (CHX) chip application in patients with peri-implant mucositis as compared to CHX gel application. **Methods:** In peri-implant sites with mucositis, CHX gel was applied in the control group (GC) and CHX chips in the test group (CC) at baseline and after three months. At baseline and after six months, peri-implant pocket depths (PPD), bleeding-on-probing (BOP) and activated matrix metalloproteinase-8 (aMMP8) were assessed. Longitudinal changes were tested for inter-group differences. **Results:** Thirty-two patients were treated. BOP was more reduced (p = 0.006) in CC than in GC, with means and standard deviations of 46 ± 28% and 17 ± 27%, respectively. PPD was more reduced (p = 0.002) in CC than in GC with 0.65 ± 0.40 mm and 0.18 ± 0.32 mm, respectively. Regarding BOP, the percentages of improved, unchanged and worsened sites accounted for 32%, 61% and 7% in GC and 46%, 53% and 1% in CC, respectively. For probing pocket depth, the according values were 26%, 66% and 8% (GC) versus 57%, 38% and 5% (CC). **Conclusions:** During supportive therapy, repeated CHX chip application might resolve marginal peri-implant inflammation in terms of bleeding better than CHX gel.

## 1. Background

Mucositis describes the first clinical manifestation of inflammation around peri-implant soft tissues, i.e., the mucosa [1,2]. Clinically, it is characterized by mucosal bleeding on peri-implant probing. This condition has been shown to be reversible in most cases, even if longer healing periods may be required [3]. Once the bony attachment is affected by inflammation and marginal bone loss is observable in radiographs or peri-implant probing proves attachment loss, the manifestation of peri-implant inflammation is called peri-implantitis. With the latter, tissue breakdown becomes irreversible [1,2].

For both forms of peri-implant disease conditions, biofilms, which are formed on the implant surfaces, are considered the primary etiologic reason for the phlogistic host response, while several risk factors have been found to modify disease initiation and progression [4,5,6]. Among these risk factors, history of periodontitis, insufficient oral hygiene and general health issues seem to constitute major problems [7,8,9] while the impact of smoking is still a matter of controversial discussion [7,10]. Like in gingivitis, which represents in many aspects, the equivalent disease around natural teeth, a careful removal and control of colonizing pathogenic bacteria allows for effective disease control [11]. The absence of BOP is considered the main surrogate parameter for the resolution of inflammation, while a laboratory test of the inflammation-triggered and activated form of matrix metalloproteinase-8 (aMMP8) has been introduced in order to describe inflammation and the degree of degradation more objectively [12]. 

Mucositis can be managed by exclusively non-surgical means. However, it seems to be more challenging to control as compared to gingivitis [13]. This fact may in part be explained by the complex abutment connection geometry and the implant neck morphology [14]. In addition, a reduced blood perfusion of the peri-implant tissues due to the missing periodontium is a further important difference between both manifestations [15].

Along with instructions for dental hygiene, biofilm removal is typically performed by mechanical means using manual or power-driven instruments with steel, carbon or resin tips [16]. In order to increase the antimicrobial effect, antiseptic agents have been suggested [17]. Most antiseptic solutions, however, have been shown to have an almost negligible effect, which has been explained by a quick clearance from the pocket due to the high crevicular fluid flow rates and excessive bleeding after treatment-related mechanical irritation [18,19,20]. Therefore, slow-release devices have been introduced, which were supposed to deliver the active antimicrobial agents within days and to keep constant and high concentration within the pockets. Chlorhexidine (CHX), a bisbiguanide antiseptic compound with a broad spectrum of antimicrobial action against oral pathogens [21], is available in a slowly dissolving chip formulation [22]. The repeated use during active and supportive periodontal therapy has already been shown to result in improved resolution of the inflammation in terms of decreased bleeding-on-probing (BOP) indices and—in some studies—in enhanced probing pocket depth reduction values [23]. However, to the best of the authors’ knowledge, no published studies are yet available, which investigate CHX chips’ application as compared to the application of CHX gel in patients with peri-implant mucositis during supportive maintenance therapy. Therefore, it was the aim of the present study to assess possible effects by repeated CHX chip application in patients with peri-implant mucositis as compared to the application of chlorhexidine gel. The hypothesis of the study was that the adjunctive use of a CHX chip would result in less inflammation as observed by a lower BOP as the primary outcome parameter. 

## 2. Methods

The study protocol was independently reviewed and approved by the liable ethics committee (North West and Central Switzerland; EKNZ 2015-175) and registered in the German registry for clinical trials (DRKS 000017782) in December 2019. The whole study was conducted in accordance to the revised declaration of Helsinki [24]. Written consent was given by each patient before inclusion. The trial was designed as a randomized prospective study with two parallel arms assessing the effect of two treatment regimes employing CHX in two different dosage forms in a private practice recall population with peri-implant mucositis in the time period from July 2015 to September 2016. Randomized allocation of the treatment groups was determined by a computerized random list (www.random.org), that was prepared by PS before inclusion of the first patient. Twenty patients with at least one implant showing peri-implant mucositis with bleeding on peri-implant probing were assigned to each treatment group. A clinical baseline examination also included an X-ray to allow for the level of the marginal peri-implant bone. The manuscript was designed adhering strictly to the consort statement for the presentation of randomized controlled studies.

### 2.1. Eligibility Criteria 

Participants conforming to the following inclusion criteria were eligible for the study:Adult women and men (≥18 year);Informed consent as documented by signature;Systemically healthy subjects;At least one dental implant loaded by fixed single crowns or bridges (screwed or cemented) with mucositis in terms of a probing pocket depth between 3 and 5 mm and bleeding on probing;Bone loss ≤ 2 mm as detectable on X-ray;Ability to read and understand the patient information.

The following criteria led to the exclusion of the participant:Patients with active periodontal pockets;Pregnant or nursing women or women planning to have a child;Patients with antibiotic therapy within the previous 6 months;Patients with peri-implantitis therapy within the previous 3 months;Subjects with an allergy to CHX;Heavy smokers (≥10 cigarettes per day);Anti-coagulated patients (exception: ≤100 mg/day acetylsalicylic acid);Removable prostheses attached on the affected implants.

Female patients willing to participate, who could not safely exclude pregnancy, were asked to wait for several weeks before inclusion, while during the study free pregnancy tests were available. 

### 2.2. Patient Recruitment, Study Group Assignment and Withdrawal

All study participants were recruited from patients in a private practice, who were in a regular dental hygiene program based on their individual periodontal and caries risk assessment. 

All patient were carefully informed in advance about the intention of the study and the related therapy by dental hygienists and CBE. Patients participated strictly on a voluntary basis. According to a previously compiled computerized randomization list and the chronological order of inclusion, each patient was assigned to either the CHX chip group (test group; CC) or the CHX gel group (control group; CG).

Group allocation was kept concealed in opaque envelopes until the first study appointment, when the concealment was broken after pocket rinsing with CHX.

Patients were withdrawn during the study by the principal investigator (CB) if they
Did not comply anymore with the inclusion criteria (i.e., intake of systemic antibiotics);Missed a study appointment;Did not comply with the study-related instructions (i.e., neglect of interproximal brushes with the regarding antiseptic agent).

Data of withdrawn patients were excluded from the analysis, and patients were not replaced. 

### 2.3. Clinical Examination 

Clinical parameters and intraoral x-rays were taken at baseline by a dental hygienist, who was specifically instructed and calibrated for implant probing, including repeated controlled-force probing at 20 g prior to the study. Clinical examination was repeated after six months. The examination included the measurement of bleeding-on-probing as primary outcome parameter and peri-implant pocket depths (PPD) and plaque index values at six sites, respectively as secondary outcome parameters [25].

For all probing depth measurements, the same type of probe was used (PCP 12, Hu-Friedy, Chicago, IL, USA).

### 2.4. Treatment

Initially, the supramucosal part of each implant was cleaned carefully with a rubber cup and polishing paste (Prophy paste RDA 40, Directa Dental, Kümmersbach, Germany) for a maximum of 5 min. Afterwards, each implant pocket was rinsed with 10 mL of chlorhexidine solution (Plakout solution 0.1%, KerrHawe SA, 6934 Bioggio, Schweiz).

In the control group (GC), a gel was applied abundantly into the sulcus (Curasept ADS 1% periodontal Gel, Curaden International AG, 6001 Kriens, Schweiz) using a blunt cannula. In the test group (CC), up to four chips (PerioChip, Karr Dental, Wollerau, Switzerland) were applied depending on the number of BOP positive sites and the depths of the regarding pocket (PPD) (Figure 1 and Figure 2). 

Patients of both groups were instructed to use interdental brushes once per day in combination with chlorhexidine gel (Curasept ADS 1% periodontal Gel, Curaden International AG, 6001 Kriens, Schweiz). The individual oral hygiene, which consisted of a manual or electric toothbrush remained unchanged, but patients were carefully reinstructed. The regarding treatment was performed by dental hygienists.

After 3 months, the above-mentioned treatment procedures at the implant sites were repeated, if bleeding persisted. Asymptomatic sites were not re-treated, i.e., only mechanical debridement and polishing was carefully performed supramucosally. 

After 6 months, the final examination including laboratory tests was performed by an investigator (CB) who was unaware of the group allocation. The measurements included BOP as primary parameter and peri-implant pocket depths and plaque index values at 6 sites as secondary outcome parameter.

### 2.5. Evaluation of aMMP-8 Levels

At baseline and after 6 months aMMP-8 samples were taken with paper tips from the entrance of the pockets. In order to avoid any falsification of the results due to eventual bleeding after probing, samples were taken prior to the microbiological test.

Sulcus fluid samples were taken after isolation of the implants with cotton rolls and cleaning of the supramucosal implant parts from the mesial site of each implant. A commercially available quantitative enzyme-linked immunosorbent assay (ELISA) for aMMP8 (Dentognostics, Matrix Biotech, Jena, Germany) was used. In the laboratory, the concentration of aMMP8 was assessed by spectroscopy. Assays were performed in duplicate and the optical density at 450 nm was determined using a microplate reader (Labsystems 125 iEMS Reader MF) with background correction at 570 nm. Readings for each sample were averaged and subtracted from the zero standard. The lower detection limits for the aMMP8 concentration was indicated as 6 pg/mL.

### 2.6. Microbiological Analysis 

Microbiological samples were also taken at baseline and after 6 months directly after aMMP8 sampling. Bacterial samples were taken with sterile paper points, which were inserted into the mesiobuccal mucosal pocket and left there for 10 s before removal. All the samples were stored in test tubes containing 100 μL of guanidine buffer and sent for microbiological analysis to the laboratory (Institut für Angewandte Immunologie IAI AG, Zuchwil, Switzerland). A polymerase chain reaction (PCR) essay based on an RNA-test (IAI Pado Test 4.5) was used to detect the bacterial concentrations. Concentrations of *Aggregatibacter actinomycetemcomitans*, *Tannerella forsythia*, *Porphyromonas gingivalis* and *Treponema denticola* were indicated in percentages (%) and the total bacterial load of the paper points (TBL) was reported as approximate count.

### 2.7. Statistical Evaluation

Statistical analyses and graphs were done with the statistical software R [26] and Microsoft Excel for Mac (Vs. 14.7.3 (170325), Microsoft Corporation, Washington, USA). In order to obtain statistically independent observations, the data of only one implant per patient, which was randomly chosen, was included in this study. Wilcoxon rank sum tests (for continuous measurement variables like BOP, PI, aMMP-8, bacterial counts and age) and Fisher’s exact tests (for count data like smoking, gender) were used to analyze differences between the CC and GC group at baseline, follow-up and Wilcoxon rank sum test with respect to the changes between baseline and follow-up. Longitudinal intra-group differences for the outcome parameters BOP, PI, aMMP-8 and bacterial counts were assessed using Wilcoxon signed rank tests. The significance level for all analyses was set to α = 0.05. 

## 3. Results

A total of 40 patients were scheduled for inclusion into the study. Two of them declined study participation prior to any study-related treatment while six patients missed study appointments due to spontaneous schedule difficulties or because they forgot their appointment, and were therefore excluded from the study. Therefore, 32 patients (15 in the GC and 17 in the CC group) completed the study-related treatment and their data were finally used for statistical analysis. 

Patient characteristics and clinical baseline data are given separately for both groups in Table 1. 

### 3.1. BOP and PPD Change

BOP was significantly more reduced (*p* = 0.006) in the group employing chips compared to the gel group, with means and standard deviations of 46 ± 28% and 17 ± 27%, respectively. 

Likewise, PPD were reduced significantly more (*p* = 0.002) in CC than in GC with 0.65 ± 0.40 mm and 0.18 ± 0.32 mm, respectively. Figure 3 and Figure 4 show box-plots depicting baseline and follow-up values for BOP and PPD.

The clinical parameters BOP and PPD are depicted for each clinical site and separately for the treatment groups before and six months after treatment (Figure 5). 

In addition, percentages of single sites that improved or worsened were calculated. With regard to bleeding, the percentages of improved, unchanged and worsened sites accounted for 32%, 61% and 7% in the GC group and 46%, 53% and 1% in the CC group, respectively, missing to show significant inter-group differences (*p* = 0.074). With regard to probing pocket depth, the values were 26%, 66% and 8% (GC) versus 57%, 38% and 5% (CC), showing significant inter-group differences (*p* = 0.001) (Figure 6). 

### 3.2. Bacterial Load (TBL) and aMMP8

Counts for total bacterial load after 6 months accounted for 19.7 ± 11.3 in the CC group and 23.1 ± 10.8 in the GC group, showing no significant differences neither for the values after six months nor for the reduction from baseline (4.8 ± 12.7 and 1.2 ± 15.3, respectively). Likewise, the analysis for single bacteria species showed no significant change (Table 2). 

Then, aMMP-8 levels after six months (21.5 ± 37.5 and 50.2 ± 147.7 ng/mL) and the respective reduction (0.5 ± 41.7 and −31.7 ± 146.4 ng/mL) did not show significant differences between the study groups.

## 4. Discussion

The present study assessed changes in the clinical and laboratory parameters six months after repeated treatment of peri-implant mucositis with CHX rinsing in combination with the application of CHX in either gel or chip form. With a stronger reduction of the mucositis in terms of BOP values in the chip group we found our hypothesis confirmed. 

This finding is in accordance with results from a randomized multi-center trial by Machtei and coworkers [27]. In that study, up to four CHX chips were placed into the pockets of ailing implants with a diagnosis of peri-implantitis. After 2, 4, 6, 8, 12 and 18 weeks, chips were replaced if pockets exceeding 5 mm were present and clinical parameters were reassessed six months after initial treatment. BOP values were significantly reduced from 100% at baseline to 58% after the intense treatment. Like in the present study, chips were applied without previous subgingival mechanical debridement. The differently severe extents of peri-implant inflammation, i.e., mucositis or peri-implantitis, are reflected in the different frequency of chip replacement, i.e., once in the present trial and five times in the cited study. 

In addition to the pronounced reduction of BOP, a better reduction of peri-implant pocket depths was observed. Since the treated sites did not show noteworthy bone loss on the baseline radiographs, the inflammation around the implant was restricted to mucosa lesions. Clinically, at the implant sites with pocket depths of up to 5 mm at baseline, slight mucosal recessions were found after six months. Both facts suggest resolution of soft tissue swelling, which is—along with bleeding—one of the classical symptoms of an inflammation. This resolution, however, might have been caused by two different ways of action: The first—and well assessed—is a reaction of the pharmacologic, i.e., antibacterial effect of chlorhexidine on bacteria in oral biofilms [28]. By controlling the antimicrobial inflammatory trigger, healthy mucosa may be re-established within an interval of days to weeks [29]. A second pathway, however, that has been noticed during the course of the present study, might be due to a mechanical irritation of the fragile mucosa due to insertion and presence of one or more of the rather rigid polylactide chips. Naturally, not perfectly adapted to the curvature of the cervical crown or the implant surface in the shoulder area, the edges of the chip might cause a local trauma on the peripheric mucosa together with mechanical stress induced by brushing or chewing. This observation is in accordance with a study by Machtei et al. (2012), who found peri-implantitis pockets not only strongly reduced if CHX chips had been repeatedly placed, but also in a control group, when CHX-free chips, consisting of polylactide only, were placed. Therefore, the reason for the observed mucosal recession might be explained by a synergism of both, a pharmaceutical and a local traumatic effect on the mucosa.

Enhanced pocket depth and BOP reduction in the present study is in accordance with results from studies that assessed non-surgical periodontitis treatment: it has been showed that CHX chips reduced pocket depths and the number of BOP positive sites in patients with chronic periodontitis better than the repeated application of CHX as solution or gel [30,31]. However, in cases which showed a markedly higher initial pocket depth and enhanced reduction as compared to the present study, CHX was applied after mechanical removal of the subgingival biofilm. 

In the present study, the findings regarding BOP as the parameter for clinical inflammation are not reflected by the values of the aMMP-8 assessment. After treatment, the concentration of aMMP-8 did not change significantly. This finding conforms with the fact that an association of aMMP-8 and BOP had actually never been shown around implants so far, whereas a correlation of the volume of sulcular fluid and aMMP-8 around implants seems to exist [12,32]. Moreover, the laboratory analysis performed in this study is technique sensitive: minute aberrations during sampling, dilution process and the final assessment of the concentration might lead to marked aberrations of the results [33]. What is more, the present study design defined aMMP-8 sampling from mesio-palatal sites before probing was performed. Accordingly, sites might have been tested that turned out to be BOP negative. Taking into consideration that peri-implant sulcus fluid, that gets soaked for the aMMP-8 analysis, is a composed product of the whole peri-implant circumference we considered the possible bias for false-negative testing to be very low. 

In this context, the reliability of the bleeding-on-probing parameter is important. Bleeding on peri-implant probing has been found even in otherwise clinically healthy implants (Salvi et al. 2012). Therefore, any probing bears a risk for false-positive recordings regarding inflammation, even though probing was performed by a well-trained and calibrated operator in the present study. Nevertheless, it is still the recommended standard test in order to clinically assess inflammation of the peri-implant mucosa (Sanz et al. 2012).

One important limitation should be considered when interpreting the study results from a clinical point of view: While supramucosal parts of the implants have been carefully polished before chips or gel were applied submucosally, no mechanical debridement of the submucosal areas has been performed. Intact biofilms, however, show a markedly higher resilience against antiseptics. It has been shown that much higher concentrations of antiseptic agents are needed to affect bacteria in a biofilm [34,35] as compared to planktonic bacteria. Accordingly, antiseptics or antibiotics show much stronger effects, if the biofilm structure is destroyed or disturbed my mechanical means [36,37]. Mechanical surface debridement, however, bears the risk of damaging the titanium surface: instrument tips of steel or titanium leave scratches [38,39] that might abet bacterial re-colonization, while softer materials like carbon or composites are prone to leave remnants of the instrument material itself on the surface. That might likewise result in disadvantageous colonization [40]. What is more, complex implant surface areas seem difficult to access by hand instrumentation [41]. Though air powder abrasives with glycine particles seem to be an alternative that is unproblematic for the titanium surface while reaching the major part of the submucosal area [42], the according device is not yet disposable in the greater part of the dental practices. Therefore, the decision for no mechanical debridement was taken for the study protocol. Interestingly, the provided therapy worked likewise very well with regard to the reduction of both inflammation and pocket depths, while the extent of healing after combined mechanical and chemical therapy is still to be assessed in future studies.

Furthermore, due to the relatively low number of sites the power of the present study is limited. Right on that account it is noteworthy, that statistical tests showed, nevertheless, difference for the BOP and PPD change between groups. 

At baseline, with 84 ± 15% significantly more sites were BOP positive in the chip group than in the gel group with 63 ± 18%. Although this has to be kept in mind as a possible bias with regard to the reduction of the parameter, a difference in baseline values is a typical finding if randomized group allocation is part of the study design. However, the significantly stronger reduction of the inflammation in the initially stronger affected group can by no means fully be explained by the initial inter-group difference only.

In accordance with a significantly improved BOP in both groups, Figure 6 shows that more peri-implant sites improved after treatment than sites that got worse. However, the clinical impact of the CHX application in form of chip or gel is discussible since in both groups most of the peri-implant sites did indeed not change. Accordingly, the overall success at six months after treatment is questionable. Long-term effects as the relevant outcome from the patient’s point of view remain unassessed in this study. Taking into consideration that the primary etiologic reason for mucositis is biofilm on the implant surface [4], hygienic morphology of the prosthetic crown and an effectively performed day-to-day oral hygiene safeguard implant health on the long-term. Once or even repeated interventions by the dentist, however, might play a secondary role. 

Furthermore, the repeated application of one or even more chips might have a relevant impact on the financial aspect of the treatment. Balancing costs and effect seems indispensable considering the rather limited amount of PPD changes shown in this study [43,44,45]. Likewise, placement of a chlorhexidine chip might cause a little more discomfort as compared to irrigation, especially if the operator is unfamiliar with this action. Nevertheless, clinical parameters showed statistically significant changes following an easy intervention. Considering the generally modest outcomes in the therapy of peri-implant diseases, the placement of CHX chips might be a useful tool in the maintenance of implants with mucositis and the avoidance of its development to peri-implantitis. 

Plaque record might seem high with scores exceeding 40 percent. O’Leary’s plaque index, however, as a dichotomous index, permits just for positive or negative recordings, urging the investigator to record positive findings strictly even if very small amounts of plaque remnants are found. That was generally the case in the present study. Within the recall program, patients had already underwent meticulous oral hygiene instructions, so that the scores have to be considered the best possible hygiene level for the individual patient.

In accordance with the study protocol, patients that did not show up to at least one study appointment were excluded from the study. Therefore, data analysis was not an intention-to-treat. Since two thirds of the regarding patients had not shown up for the very first appointment and concealment was not even broken the potential bias was considered low and these patients were excluded from any analysis.

Smokers were not excluded from the study. With three smoking patients in the gel group and only one in the chip group, the patient collective showed an unequal distribution that was found not to reach statistical difference. Since smoking has been shown to have a suppressive effect on BOP values [46,47] and less favorable results after periodontal therapy [48,49] the results of the present study should be interpreted with care in this regard, even though few smokers were involved.

In the present study, two treatment modalities both including the use of CHX have been compared. From a theoretical point of view a comparison with a negative control without the use of any antiseptic agent would have been reasonable. Likewise, the different composition and adjuvants of the CHX gel (including the antidiscolorant device) and the chip (with a gelatin carrier), containing also different concentrations of the active agent, might be critical for a basic research approach. From a practical point of view, however, the additional benefit of a chip as a slow delivery device as compared to a most usual treatment modality [29] is of special interest.

## 5. Conclusions

Application of the CHX chip may reduce the number of bleeding sites around implants with mucositis, and a reduction in probing pocket depths was found in the setting of the present study. Therefore, its worth as a tool in a preventive protocol for peri-implantitis might be considered. 

## 6. Declarations

### 6.1. Ethics Approval and Consent to Participate 

The study was performed according the approved study application by the ethics committee of North West and Central Switzerland (EKNZ 2015-175). Written consent was given by each patient for participation and later publication. 

### 6.2. Consent for Publication

Written consent was given by each patient for participation and later publication.

### 6.3. Availability of Data and Material

Datasets for this study are available from the study PI on reasonable request. 

## Figures and Tables

**Figure 1 dentistry-07-00115-f001:**
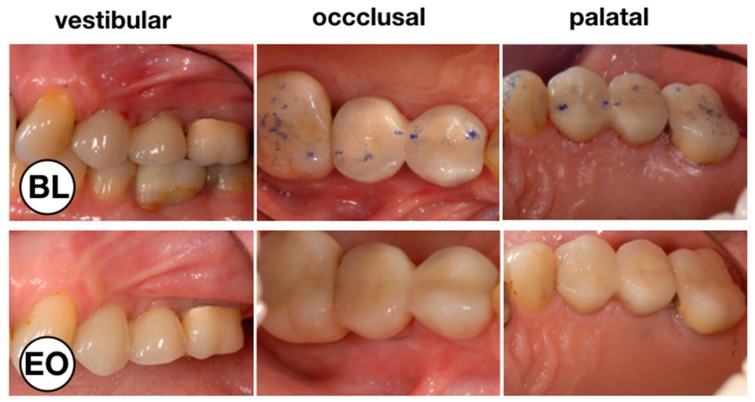
Clinical situation before (BL) and at the end of the observation period (EO) after the test treatment with chlorhexidine (CHX) chips. Vestibular, occlusal and palatal aspect of implant 24 with peri-implant mucositis.

**Figure 2 dentistry-07-00115-f002:**
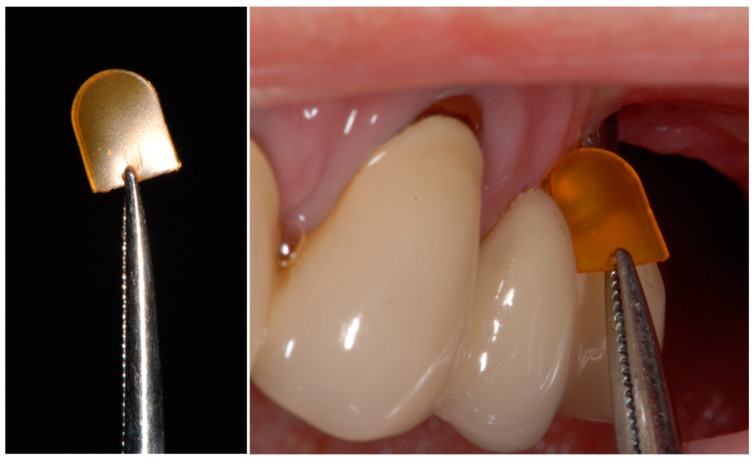
Placement of a CHC chip. A firm hold of the CHX chip with dental tweezers (**left**) in order to safely insert the chip into the peri-implant sulcus (**right**).

**Figure 3 dentistry-07-00115-f003:**
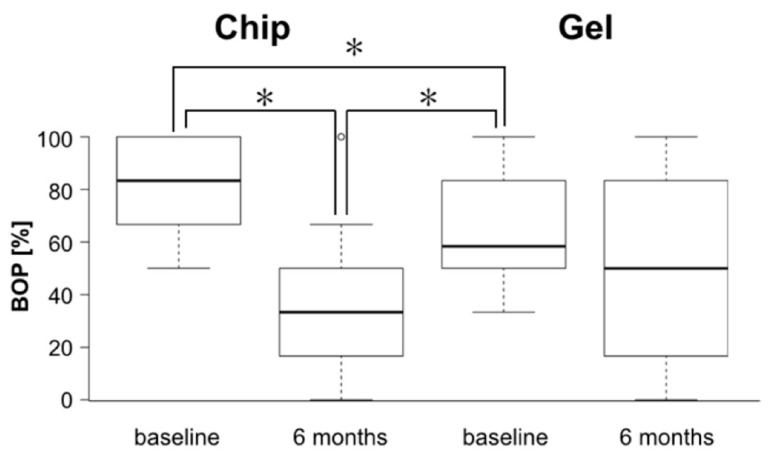
Box-plots for bleeding-on-probing (BOP) values at baseline and after six months for the chip and the gel group, respectively. Boxes indicate interquartile ranges, horizontal bars median values, the whiskers 1.5 * interquartile range or minima/maxima of the distribution and the unfilled dot an outlier in the chip-employing group. The asterisks indicate statistically significant differences.

**Figure 4 dentistry-07-00115-f004:**
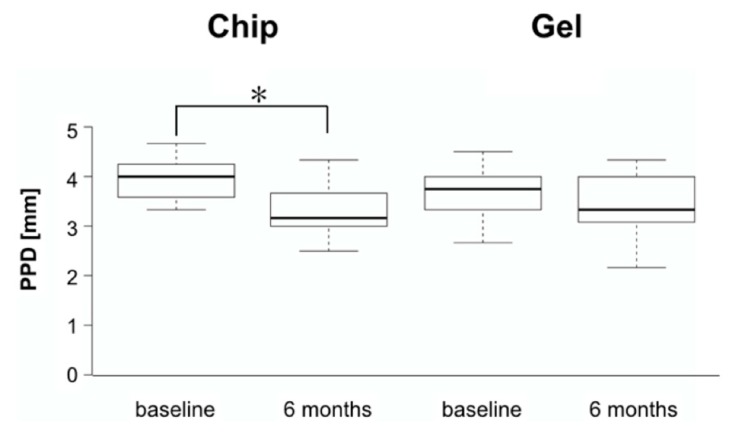
Boxplots of peri-implant pocket depths (PPD) at baseline and after six months for the chip and the gel group, respectively. Boxes indicate interquartile ranges, horizontal bars median values and the whiskers 1.5 * interquartile range or minima/maxima of the distribution. The asterisk indicates a statistically significant difference.

**Figure 5 dentistry-07-00115-f005:**
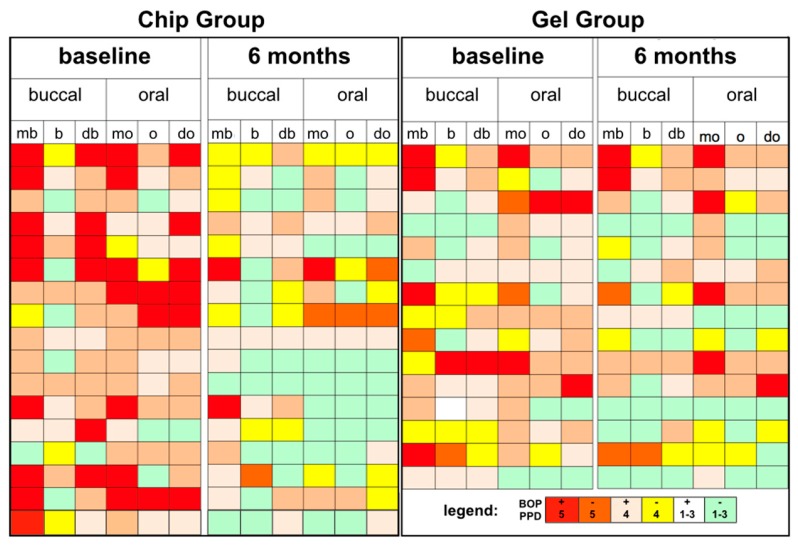
Site-specific report of the chip group (n = 17) and gel group (n = 15) with regard to bleeding (BOP) and probing pocket depths (PPD) status. mb—mesiobuccal, b—buccal, db—distobuccal, mo—mesio-oral, o—oral, do—disto-oral.

**Figure 6 dentistry-07-00115-f006:**
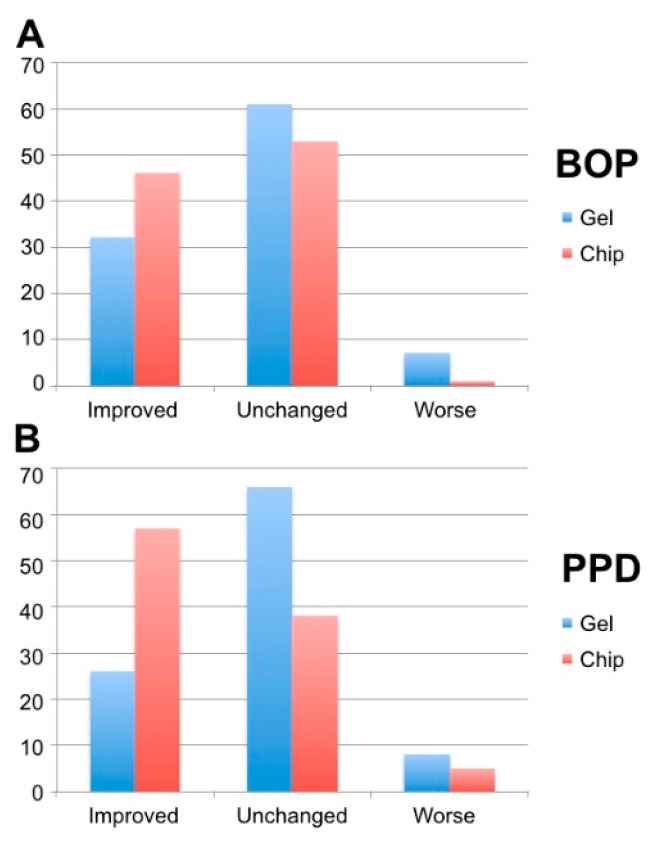
Percentages of improved, unchanged and worsened sites regarding. Bleeding-on-probing (BOP) and peri-implant pocket depths (PPD).

**Table 1 dentistry-07-00115-t001:** Baseline characteristics of the patients in the CHX chip and the CHX gel group, respectively.

**Patient Characteristics**	**CHX Chip**	**CHX Gel**	***p***
Patients	17	15	
Gender (m/f)	9/8	7/8	1 ^1^
Age (year)	60.0	57.5	0.5 ^2^
Smokers (n)	1	3	0.3 ^1^
Implant position ul/ua/ll/la (n)	5/7/7/1	8/3/5/0	0.341 ^3^
Implant loading time (year)	5.6 ± 1.6	5.5 ± 1.8	0.67 ^4^
Total number of implants Per patient (mean ± std %)	2.7 ± 2.0	2.8 ± 2.1	0.83 ^2^
**Clinical parameters**			
BOP (mean ± std %)	84 ± 15	63 ± 18	**0.003** ^2^
PDD (mean ± std mm)	3.9 ± 0.5	3.7 ± 0.6	0.1 ^2^
PI (mean ± std %)	42.2 ± 37	45.6 ± 34	0.8 ^2^
**Laboratory parameters**			
TBL (means ± std)	24.7 ± 11	23.9 ± 14.2	1.0 ^2^
aMMP8 (means ± std)	21.8 ± 21.5	17.3 ± 10.1	0.5 ^2^

CC—CHX Chip group, GC—CHX Gel group, ul—upper laterals, ua—upper anteriors, ll—lower laterals, la—lower anteriors. ^1^ Fisher’s exact test; ^2^ Kruskal–Wallis test; ^3^ Pearson’s chi square test; ^4^ Mann–Whitney-Test; bold *p*-values indicate a statistically significant difference between the groups; BOP—bleeding on probing, PPD—peri-implant probing depth, PI—plaque index, TBL—total bacterial load, aMMP-8—activated matrix metalloproteinase 8.

**Table 2 dentistry-07-00115-t002:** Percentual bacteria counts for the single strains tested at baseline and the end of observation after six months.

Bacterial Species	Baseline		End of Observation	
	GC	CC	GC	CC
*A. actinomycetemcomitans*	0.0 + 0.0	0.0 + 0.0	0.0014 ± 0.005	0.0 + 0.0
*T. forsythia*	0.197 ± 0.395	0.024 ± 0.395	0.033 ± 0.129	0.15 ± 0.364
*P. gingivalis*	0.0007 + 0.0026	0.0 + 0.0	0.013 ± 0.044	0.012 ± 0.051
*T. denticola*	0.136 + 0.0398	0.126 ± 0.038	0.051 ± 0.180	0.07 ± 0.221

No statistically significant changes were found, neither comparing baseline and end values by the Wilcoxon test nor comparing the treatment groups using Mann–Whitney test, respectively.

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
