# Peer review of "Treatment of Peri-Implant Mucositis with Repeated Application of Chlorhexidine Chips or Gel during Supportive Therapy—A Randomized Clinical Trial"

_dentistry, 2019, doi:10.3390/dj7040115_

Round 1

Reviewer 1 Report

This manuscript, aimed to compare the chlorhexidine chip application VS the chlorhexidine gel application in peri-implant mucositis, is well organized and the paper appears well written.

The topic is of great interest and the study design is properly planned.

Despite the small sample size, it could be interesting to describe the patients' degree of satisfaction. 

Furthermore, some clinical pictures related to the treatment could add relevance to the study.

Author Response

(Please download attached document to see the sections changed in the revised manuscript)

Dear Professor Jaquiéry

We would like to thank you for the editorial comments and the reviewers for the careful revision of the present study entitled

Treatment of Peri-implant Mucositis with Repeated Application of Chlorhexidine Chips or Gel during Supportive Therapy – A Randomized Clinical Trial

by Philipp Sahrmann, Cyrill Bettschart, Daniel B. Wiedemeier, Ahmed Al-Majid, Thomas Attin and Patrick R. Schmidlin

The appreciation for the presented study and the constructive criticism were highly estimated.

In the following, we will respond point by point to the reviewers’ suggestions and questions and indicate the text sections with the regarding changes that we performed in the revised version.

In the revised text, we have been highlighting all changes in the track-change mode.

We feel that important points have been addressed thereby and the quality of the paper was improved.

With best regards from Zurich,

Philipp Sahrmann

Reviewer 1:

Comment 1

This manuscript, aimed to compare the chlorhexidine chip application VS the chlorhexidine gel application in peri-implant mucositis, is well organized and the paper appears well written.

The topic is of great interest and the study design is properly planned.

            Authors’ response: The authors want to thank the referee for the kind appreciation!

            Changed text section:

            (na)

Comment 2:

Despite the small sample size, it could be interesting to describe the patients' degree of satisfaction. 

Authors’ response: The authors want to thank the referee for the constructive criticism. Since data collection is closed it is hardly possible to assess according data in a concordance with study design and ethical approval (readdressing patients). However, we understand that the potential benefit of the test treatment has to be put into the context of its disadvantages. Accordingly, we added a paragraph to the revised discussion section, in which we discuss the issue.

            Changed text section:

Furthermore, the repeated application of one or even more chips might have a relevant impact on the financial aspect of the treatment. Balancing costs and effect seems indispensable considering the rather limited amount of PPD changes shown in this study [41–43]. Likewise, placement of a chlorhexidine chip might cause a little more discomfort as compared to irrigation, especially if the operator is unfamiliar with this action. Nevertheless, clinical parameters showed statistically significant changes following an easy intervention.

Comment 3:

Furthermore, some clinical pictures related to the treatment could add relevance to the study.

Authors’ response: Again, the authors appreciate the input of the referee and we would like to thank for the possibility to illustrate our findings by adding a picture depicting a clinical case at baseline and at the end of the observation period.

            Changed text section:

Figure 1   Clinical situation before (BL) and at the end of the observation period (EO) after the test treatment with CHX chips

Vestibular, occlusal and palatal aspect of implant 24 with peri-implant mucositis.

Reviewer 2 Report

Reviewer Comments to Author: 
Dear authors, 
I appreciate the efforts of the study with the title "Treatment of Peri-implant Mucositis with Repeated Application of Chlorhexidine Chips or Gel during Supportive Therapy – A Randomized Clinical Trial". In principle, the topic of this investigation is of potential clinical interest. The manuscript is well written, the structure is clear. I just have some minor comments.

-Figures: please provide some clinical pictures if possible, e.g. a picture of the test group at baseline and after six months that underline the findings of this paper, an image of the chip application would be good for a better understanding

-M&M: How exactly was the sulcus fluid sample of the MMP8 investigation taken, was any quantification of the fluid volume applied? It is necessary to analyze MMP8 level in relation to the sulcus fluid volume.

-Results: -Add the exact results of the single bacteria species evaluation (line 237)

                -Please add the unit to the MMP8 results, pg/ml? What was the exact total    

                 bacterial count, is it cell count or CFU?

                -How long were the implants in function and what are the implant positions?

Author Response

(Please download attached document to see the sections changed in the revised manuscript)

Dear Professor Jaquiéry

We would like to thank you for the editorial comments and the reviewers for the careful revision of the present study entitled

Treatment of Peri-implant Mucositis with Repeated Application of Chlorhexidine Chips or Gel during Supportive Therapy – A Randomized Clinical Trial

by Philipp Sahrmann, Cyrill Bettschart, Daniel B. Wiedemeier, Ahmed Al-Majid, Thomas Attin and Patrick R. Schmidlin

The appreciation for the presented study and the constructive criticism were highly estimated.

In the following, we will respond point by point to the reviewers’ suggestions and questions and indicate the text sections with the regarding changes that we performed in the revised version.

In the revised text, we have been highlighting all changes in the track-change mode.

We feel that important points have been addressed thereby and the quality of the paper was improved.

With best regards from Zurich,

Philipp Sahrmann

Reviewer 2

Comment 1

Dear authors, 
I appreciate the efforts of the study with the title "Treatment of Peri-implant Mucositis with Repeated Application of Chlorhexidine Chips or Gel during Supportive Therapy – A Randomized Clinical Trial". In principle, the topic of this investigation is of potential clinical interest. The manuscript is well written, the structure is clear. I just have some minor comments.

Authors’ response: The authors want to thank the referee for the appreciation and – in the following - helpful and constructive criticism.

            Changed text section:

(na)

Comment 2:
Figures: please provide some clinical pictures if possible, e.g. a picture of the test group at baseline and after six months that underline the findings of this paper, an image of the chip application would be good for a better understanding

Authors’ response: Additional figures (new Fig. 1 and 2) were added in order to better illustrate the treatment with the chx chips

            Changed text section:

Figure 1   Clinical situation before (BL) and at the end of the observation period (EO) after the test treatment with CHX chips

Vestibular, occlusal and palatal aspect of implant 24 with peri-implant mucositis.

Figure 2   Placement of a chx chip

A firm hold of the chx chip with dental tweezers (left) in order to safely insert the chip into the peri-implant sulcus (right).

Comment 3:
M&M: How exactly was the sulcus fluid sample of the MMP8 investigation taken, was any quantification of the fluid volume applied? It is necessary to analyze MMP8 level in relation to the sulcus fluid volume.

Authors’ response: The sampling for MMP8 had been described in the section “Evaluation of aMMP-8 levels”: The samples were taken with paper tips from the isolated entrance of the peri-implant sulcus. In order to avoid bleeding the respective sample was taken before microbiologic sampling (from the depth of the sulcus). With regard to the quantification, the assessment of aMMP-8 in the present study was made based on concentration levels. This was done to facilitate comparison with previously publishes studies on aMMP-8 and periodontitis or peri-implantitis (Munjal SK et al. 2007, Janska E et al. 2016, Thierbach R et al. 2016). Like in the cited studies, sulcus fluid volume has not been assessed separately since the additional information (i.e. calculation of the total amount of aMMP-8) is an interesting issue for answering further questions in baseline research on one hand, but was not focus of the present study on the other hand.

            Changed text section:

(na)

Comment 4:

Results: -Add the exact results of the single bacteria species evaluation (line 237)

            Authors’ response: The requested information was added in a new table 2.

            Changed text section:

Table 2          Percentual bacteria counts for the single strains tested at baseline and the end of observation after 6 months

Baseline

End of observation

GC

CC

GC

CC

A. actinomycetemcomitans

0.0+0.0

0.0+0.0

0.0014±0.005

0.0+0.0

T. forsythia

0.197±0.395

0.024±0.395

0.033±0.129

0.15±0.364

P. gingivalis

0.0007+0.0026

0.0+0.0

0.013±0.044

0.012±0.051

T.a denticola

0.136+0.0398

0.126±0.038

0.051±0.180

0.07±0.221

No statistically significant changes were found, neither comparing baseline and end values by Wilcoxon test nor comparing the treatment groups using Mann Whitney test, respectively.

Comment 5:

-Please add the unit to the MMP8 results, pg/ml?

            Authors’ response: Authors’ response: Thank you for the suggesting to add the unit - what was done in the revised manuscript.

            Changed text section:

Then, aMMP-8 levels after six months (21.5 ± 37.5 and 50.2 ± 147.7 ng/ml) and the respective reduction (0.5 ± 41.7 and -31.7 ± 146.4 ng/ml) did not show significant differences between the study groups.

Comment 6:

What was the exact total bacterial count, is it cell count or CFU?

Authors’ response: In our manuscript with TBL (total bacterial load) we indicated the parameter given by the commercial laboratory. The respective value relies on RNA amplification and thereby estimates cell counts but cannot give an exact number, of course.

Changed text section: (na)

Comment 7:

How long were the implants in function and what are the implant positions?

Authors’ response: Information regarding both the implant loading time and the implant position (divided in upper and lower laterals and anteriors) was added to table 1.

            Changed text section:

Table 1          Baseline characteristics of the patients in the CHX chip and the CHX                gel group, respectively

CHX chip

CHX gel

p

Patient characteristics

Patients

Gender (m/f)

Age [y]

Smokers [n]

Implant position ul/ua/ll/la [n]

Implant loading time [y]

Total number of implants

Per patient [mean±std %]

17

9/8

60.0

1

5/7/7/1

5.6±1.6

2.7 ± 2.0

15

7/8

57.5

3

8/3/5/0

5.5±1.8

2.8 ± 2.1

11

0.52

0.31

0.3413

0.674

0.832

Clinical parameters

BOP [mean±std %]

PDD [mean±std mm]

PI [mean±std %]

84 ± 15

3.9 ± 0.5

42.2 ± 37

63 ± 18

3.7 ± 0.6

45.6 ± 34

0.0032

0.12

0.82

Laboratory parameters

TBL [means±std]

aMMP8 [means±std]

24.7 ± 11

21.8 ± 21.5

23.9±14.2

17.3±10.1

1.02

0.52

CC – CHX Chip group, GC – CHX Gel group, ul – upper laterals, ua – upper anteriors, ll – lower laterals, la – lower anteriors

1 Fisher’s exact test

2 Kruskal-Wallis test

3 Pearson’s chi square test

4 Mann-Whitney-Test

bold p-values indicate a statistically significant difference between the groups

BOP – Bleeding on probing, PPD – Peri-implant probing depth, PI – Plaque index, TBL – Total bacterial load, aMMP-8 – Activated matrix metalloproteinase 8

Reviewer 3 Report

Comments to the Author(s):

Dear Author,

In your manuscript You investigate effect of chlorhexidine (CHX) chip application versus  CHX gel application in patients with 21 peri-implant mucositis. You found better effetc with CHX chip vs CHX gel application on Bleeding.

IT’s a good work but some interventions are need.

Abstructs

Regarding BOP, the percentages of improved,

29 unchanged and worsened sites accounted for 32%, 61% and 7% in GC and 46%, 53% and 1% in CC,

30 respectively. For probing pocket depth, the according values were 26%, 66% and 8% (GC) versus

31 57%, 38% and 5% (CC).

The different data found in percentages are statistically different or not?

In the introduction,

In  the following  sentence :

For both forms of peri-implant disease conditions, biofilms, which are formed on the implant

44 surfaces, are considered the primary etiologic reason for the phlogistic host response, while several

45 risk factors have been found to modify disease initiation and progression [4, …*].

please  added the References :

*Lucarini G, Zizzi A, Rubini C, Ciolino F, Aspriello SD. VEGF, Microvessel Density, and CD44 as Inflammation Markers in Peri-implant Healthy Mucosa, Peri-implant Mucositis, and Peri-implantitis: Impact of Age, Smoking, PPD, and Obesity. Inflammation. 2019 Apr;42(2):682-689. doi: 10.1007/s10753-018-0926-0.

Zizzi A, Aspriello SD, Rubini C, Goteri G. Peri-implant diseases and host inflammatory response involving mast cells: a review. Int J Immunopathol Pharmacol. 2011 Jul-Sep;24(3):557-66. Review.)

Delate because It’s not necessary:

Most antiseptic

62 solutions, however, have been shown to have an almost negligible effect, which has been explained

63 by a quick clearance from the pocket due to the high crevicular fluid flow rates and excessive bleeding

64 after treatment-related mechanical irritation [16–18].

Therefore, slow-release devices have been

65 introduced, which were supposed to deliver the active antimicrobial agents within days and to keep

66 constant and high concentration within the pockets. Chlorhexidine (CHX), a bisbiguanide antiseptic

67 compound with a broad spectrum of antimicrobial action against oral pathogens [19], is available in

68 a slowly dissolving chip formulation [20].

Change with this sentence:

Chlorhexidine (CHX), a bisbiguanide antiseptic

67 compound with a broad spectrum of antimicrobial action against oral pathogens [19], is available in

68 a slowly dissolving chip formulation [20] wich is supposed to deliver the active antimicrobial agents within days and to keep constant and high concentration within the pockets.

In Material and Method

Treatment

In the control group (GC), a gel was applied abundantly 142 into the sulcus (Curasept ADS 1%

143 periodontal Gel, Curaden International AG, 6001 Kriens, Schweiz) using a blunt cannula. In the test

144 group (CC), up to four chips (PerioChip, Karr Dental, Wollerau, Switzerland) were applied

145 depending on the number of BOP positive sites and the depths of the regarding pocket (PPD).

What are the excipients / carriers of the chip or gel? What is the percentage of PerioChip?

 Could they influence the results? Have you considered this possibility?

Another important question is that CHX gel  contain ADS system and CHX chip no. Why  didn't you use a gel CHX without ADS?

Could ADS system  reduce antimicrobial power of CHX?!

See: J Investig Clin Dent. 2014 Feb;5(1):15-22. doi: 10.1111/jicd.12050. Epub 2013 Jun 14. Evaluation of the antigingivitis effect of a chlorhexidine mouthwash with or without an antidiscoloration system compared to placebo during experimental gingivitis. Li W1, Wang RE, Finger M, Lang NP.

Discussion

Smokers were not excluded from the study. With three smoking patients in the gel group and

353 only 1 in the chip group the patient collective showed an unequal distribution that was found not to

354 reach statistical difference. Since smoking has been shown to have a suppresive effect on BOP values

355 [44; 45] and less favourable results after periodontal therapy [46; 47] the results of the present study

356 should be interpreted with care in this regard, even though few smokers were envolved.

Delete references 44 and 46

Conclusion

Delete

However, based on the effects of the pharmacological effect of CHX slutions

362 in periodontitis the influence of conventional topical CHX might be neligible [48].

References

There are several recent publications that you could use to enrich the discussion.

For example:

A Topical Desiccant Agent in Association with Manual Debridement in the Initial Treatment of Peri-Implant Mucositis: A Clinical and Microbiological Pilot Study. Lombardo G, Signoriello A, Corrocher G, Signoretto C, Burlacchini G, Pardo A, Nocini PF. Antibiotics (Basel). 2019 Jun 18;8(2). pii: E82. doi: 10.3390/antibiotics8020082.

Author Response

(Please download attached document to see the sections changed in the revised manuscript)

Dear Professor Jaquiéry

We would like to thank you for the editorial comments and the reviewers for the careful revision of the present study entitled

Treatment of Peri-implant Mucositis with Repeated Application of Chlorhexidine Chips or Gel during Supportive Therapy – A Randomized Clinical Trial

by Philipp Sahrmann, Cyrill Bettschart, Daniel B. Wiedemeier, Ahmed Al-Majid, Thomas Attin and Patrick R. Schmidlin

The appreciation for the presented study and the constructive criticism were highly estimated.

In the following, we will respond point by point to the reviewers’ suggestions and questions and indicate the text sections with the regarding changes that we performed in the revised version.

In the revised text, we have been highlighting all changes in the track-change mode.

We feel that important points have been addressed thereby and the quality of the paper was improved.

With best regards from Zurich,

Philipp Sahrmann

Reviewer 3

Comment 1

Dear Author,

In your manuscript You investigate effect of chlorhexidine (CHX) chip application versus  CHX gel application in patients with 21 peri-implant mucositis. You found better effetc with CHX chip vs CHX gel application on Bleeding.

IT’s a good work but some interventions are need.

Authors’ response: The authors want to thank the reviewer for the diligent and constructive suggestions and criticism.

            Changed text section:

(na)

Comment 2

Abstructs

Regarding BOP, the percentages of improved, unchanged and worsened sites accounted for 32%, 61% and 7% in GC and 46%, 53% and 1% in CC, respectively. For probing pocket depth, the according values were 26%, 66% and 8% (GC) versus 57%, 38% and 5% (CC).

The different data found in percentages are statistically different or not?

Authors’ response: The authors want to thank the referee for an interesting question. Since we totally agree that the respective information is needed we listed the p-values in the revised result section. Since the chi square tests for the composed data is less specific we abstained to list in the abstract in order not to worsen the legibility of the necessarily tight abstract text.

            Changed text section:

With regard to bleeding, the percentages of improved, unchanged and worsened sites accounted for 32%, 61% and 7% in the GC group and 46%, 53% and 1% in the CC group, respectively, missing to show significant inter-group differences (p = 0.074). With regard to probing pocket depth, the values were 26%, 66% and 8% (GC) versus 57%, 38% and 5% (CC), showing significant inter-group differences (p = 0.001) (Fig. 6).

Comment 3

In the introduction, in  the following  sentence :

For both forms of peri-implant disease conditions, biofilms, which are formed on the implant surfaces, are considered the primary etiologic reason for the phlogistic host response, while several risk factors have been found to modify disease initiation and progression [4, …*].

please  added the References :

*Lucarini G, Zizzi A, Rubini C, Ciolino F, Aspriello SD. VEGF, Microvessel Density, and CD44 as Inflammation Markers in Peri-implant Healthy Mucosa, Peri-implant Mucositis, and Peri-implantitis: Impact of Age, Smoking, PPD, and Obesity. Inflammation. 2019 Apr;42(2):682-689. doi: 10.1007/s10753-018-0926-0.

Zizzi A, Aspriello SD, Rubini C, Goteri G. Peri-implant diseases and host inflammatory response involving mast cells: a review. Int J Immunopathol Pharmacol. 2011 Jul-Sep;24(3):557-66. Review.)

Authors’ response:

The authors checked the suggested references and were happy to add both references.

Changed text section:

For both forms of peri-implant disease conditions, biofilms, which are formed on the implant surfaces, are considered the primary etiologic reason for the phlogistic host response, while several risk factors have been found to modify disease initiation and progression [4–6].

Comment 4:

Delate because It’s not necessary:

Most antiseptic solutions, however, have been shown to have an almost negligible effect, which has been explained by a quick clearance from the pocket due to the high crevicular fluid flow rates and excessive bleeding after treatment-related mechanical irritation [16–18].

Therefore, slow-release devices have been introduced, which were supposed to deliver the active antimicrobial agents within days and to keep constant and high concentration within the pockets. Chlorhexidine (CHX), a bisbiguanide antiseptic compound with a broad spectrum of antimicrobial action against oral pathogens [19], is available in a slowly dissolving chip formulation [20].

Change with this sentence:

Chlorhexidine (CHX), a bisbiguanide antiseptic compound with a broad spectrum of antimicrobial action against oral pathogens [19], is available in a slowly dissolving chip formulation [20] wich is supposed to deliver the active antimicrobial agents within days and to keep constant and high concentration within the pockets.

Authors’ response: The authors want to thank for this suggestion and we understand the referee’s intention to keep the introduction as short as possible. Since we have noticed in the past that clearance as one of the most reasonable reasons for concentration loss is often misunderstood or underestimated, we decided to leave the text section unchanged. We hope that the referee might accept.

Changed text section:

(na)

Comment 5:

In Material and Method

Treatment

In the control group (GC), a gel was applied abundantly 142 into the sulcus (Curasept ADS 1% periodontal Gel, Curaden International AG, 6001 Kriens, Schweiz) using a blunt cannula. In the test group (CC), up to four chips (PerioChip, Karr Dental, Wollerau, Switzerland) were applied depending on the number of BOP positive sites and the depths of the regarding pocket (PPD).

 What are the excipients / carriers of the chip or gel? What is the percentage of PerioChip?

 Could they influence the results? Have you considered this possibility?

Authors’ response: Gel and chip are different supplement carrying the same active component chx, but in different concentrations and different adjuvants that characterize their pharmacokinetics. Therefore, the 2.5 mg of chx within the gelatin matrix of the chip have been shown to maintain a comparatively high concentration of chx in the sulcis fluid (see also Soskolne WA et al. 1998).

With the rather comprehensive introduction we took any respective discussion for granted and concentrated on the clinical effect. However, we decided to add a paragraph to the discussion, addressing the point raised by the referee:

Changed text section:

From a theoretical point of view a comparison with a negative control without the use of any antiseptic agent would have been reasonable. Likewise, the different composition and adjuvants of the chx gel (including the antidiscolorant device) and the chip (with a gelatin carrier), containing also different concentrations of the active agent, might be critical for a basic research approach. From a practical point of view, however, the additional benefit of a chip as a slow delivery device as compared to a most usual treatment modality [27]

Comment 6:

Another important question is that CHX gel  contain ADS system and CHX chip no. Why  didn't you use a gel CHX without ADS?

Could ADS system  reduce antimicrobial power of CHX?!

See: J Investig Clin Dent. 2014 Feb;5(1):15-22. doi: 10.1111/jicd.12050. Epub 2013 Jun 14. Evaluation of the antigingivitis effect of a chlorhexidine mouthwash with or without an antidiscoloration system compared to placebo during experimental gingivitis. Li W1, Wang RE, Finger M, Lang NP.

Authors’ response: Again, the authors thank for the valuable input. And – of course, the referee is right with the suggestion regarding AHD in the gel.

The respective gel, however is one of the most often chosen products by dental practitioners when aiming to benefit from antiseptic effects and likewise avoiding trouble with the patients due to stains on the affected teeth. Again, the aim of the study was to compare from a practical point of view to similar regiments, with an already accepted control treatment on one hand an a potential alternative on the other. Accordingly we addressed the ADH issue in the section, that was changed in the revised manuscript:

Changed text section:

Likewise, the different composition and adjuvants of the chx gel (including the antidiscolorant device) and the chip (with a gelatin carrier), containing also different concentrations of the active agent, might be critical for a basic research approach.

 Comment 8:

Discussion

Smokers were not excluded from the study. With three smoking patients in the gel group and only 1 in the chip group the patient collective showed an unequal distribution that was found not to reach statistical difference. Since smoking has been shown to have a suppresive effect on BOP values [44; 45] and less favourable results after periodontal therapy [46; 47] the results of the present study should be interpreted with care in this regard, even though few smokers were envolved.

Delete references 44 and 46

Authors’ response: The authors re-checked the respective references. With all respect for the suggestion we could not figure out why the references were wrong in any way. Therefore we decided to still list them and hope that is ok for the referee. If not we kindly ask the referee to give us a hand with provision of respective reasons.

Changed text section:

(na)

Comment 9:

Conclusion

Delete

However, based on the effects of the pharmacological effect of CHX solutions in periodontitis the influence of conventional topical CHX might be neligible [48].

Authors’ response: The authors have critically discussed this suggestion and finally, we agreed to eliminate the sentence.

Changed text section:

(sentence removed in the revised manuscript)

Comment 10:

References

There are several recent publications that you could use to enrich the discussion.

For example:

A Topical Desiccant Agent in Association with Manual Debridement in the Initial Treatment of Peri-Implant Mucositis: A Clinical and Microbiological Pilot Study. Lombardo G, Signoriello A, Corrocher G, Signoretto C, Burlacchini G, Pardo A, Nocini PF. Antibiotics (Basel). 2019 Jun 18;8(2). pii: E82. doi: 10.3390/antibiotics8020082.

Authors’ response: The referee is right: The topic of peri-implant mucositis and peri-implantitis is a very dynamic field and a lot of new and interesting approaches have been proposed. The original version of the discussion section had therefor been much more comprehensive, and finally we had to eliminate many references and issues that were even stronger connected to the main issue of the present study. We therefore decided not to re-amplify our discussion with an - though interesting – reference, that is nevertheless a bit out of the focus.

Changed text section:

(na)
